# Simultaneous Study of Anti-Ferroptosis and Antioxidant Mechanisms of Butein and (*S*)-Butin

**DOI:** 10.3390/molecules25030674

**Published:** 2020-02-05

**Authors:** Jie Liu, Xican Li, Rongxin Cai, Ziwei Ren, Aizhen Zhang, Fangdan Deng, Dongfeng Chen

**Affiliations:** 1School of Basic Medical Science, Guangzhou University of Chinese Medicine, Guangzhou 510006, China; 15014173165@163.com; 2The Research Center of Basic Integrative Medicine, Guangzhou University of Chinese Medicine, Guangzhou 510006, China; 3School of Chinese Herbal Medicine, Guangzhou University of Chinese Medicine, Waihuan East Road No. 232, Guangzhou Higher Education Mega Center, Guangzhou 510006, China; choi_roy@foxmail.com (R.C.); 939725358_asd@sina.com (Z.R.); emotionfire@sina.com (A.Z.); fangdandeng1@163.com (F.D.)

**Keywords:** butein, (*S*)-butin, anti-ferroptosis, antioxidant, 2′-hydroxy chalcone, isomerization

## Abstract

To elucidate the mechanism of anti-ferroptosis and examine structural optimization in natural phenolics, cellular and chemical assays were performed with 2′-hydroxy chalcone butein and dihydroflavone (*S*)-butin. C11-BODIPY staining and flow cytometric assays suggest that butein more effectively inhibits ferroptosis in erastin-treated bone marrow-derived mesenchymal stem cells than (*S*)-butin. Butein also exhibited higher antioxidant percentages than (*S*)-butin in five antioxidant assays: linoleic acid emulsion assay, Fe^3+^-reducing antioxidant power assay, Cu^2+^-reducing antioxidant power assay, 2-phenyl-4,4,5,5-tetramethylimidazoline-1-oxyl 3-oxide radical (PTIO^•^)-trapping assay, and α,α-diphenyl-β-picrylhydrazyl radical (DPPH^•^)-trapping assay. Their reaction products with DPPH^•^ were further analyzed using ultra-performance liquid chromatography coupled with electrospray ionization quadrupole time-of-flight tandem mass spectrometry (UPLC-ESI-Q-TOF-MS). Butein and (*S*)-butin produced a butein 5,5-dimer (*m/z* 542, 271, 253, 225, 135, and 91) and a (*S*)-butin 5′,5′-dimer (*m/z* 542, 389, 269, 253, and 151), respectively. Interestingly, butein forms a cross dimer with (*S*)-butin (*m/z* 542, 523, 433, 419, 415, 406, and 375). Therefore, we conclude that butein and (*S*)-butin exert anti-ferroptotic action via an antioxidant pathway (especially the hydrogen atom transfer pathway). Following this pathway, butein and (*S*)-butin yield both self-dimers and cross dimers. Butein displays superior antioxidant or anti-ferroptosis action to (*S*)-butin. This can be attributed the decrease in π-π conjugation in butein due to saturation of its α,β-double bond and loss of its 2′-hydroxy group upon biocatalytical isomerization.

## 1. Introduction

Ferroptosis is a form of cell death activated by iron oxidation [1,2]. Ferroptosis regulation offers a new strategy for the treatment of various diseases, including cancer [3,4] and Alzheimer’s disease [5,6]. Recently, ferroptosis inhibitors have been reported, including dietary flavones, such as baicalein [5], that are bio-synthesized from dietary chalcones (especially 2′-OH chalcone) [7]. Our research suggests that the biosynthesis of flavone from 2′-OH chalcone is an antioxidant reduction process [8]. As a result, we postulate that 2′-OH chalcone will be a more effective ferroptosis inhibitor compared to its corresponding flavone. However, no direct evidence has been reported previously.

To confirm our postulation, 2′-OH chalcone butein was selected as a model compound for this study. Chalcone synthase (CHS) usually imposes a 2′-OH (2′-hydroxy group) to the *A*-ring [7,9], thus most chalcones contain a 2′-OH [10,11]. In addition, its isomer, (*S*)-butin, was investigated in the study.

As shown in Figure 1, the name (*S*)-butin refers to (2*S*)-butin (i.e., (*−*)-butin). CHI-mediated isomerization selectively produces a *2S* (rather than *2R*) absolute configuration [7,9]; thus, (*2S*) dihydroflavone predominates the dihydroflavones for which the configuration is known [10,11]. Structurally, both (*S*)-butin and butein possess the chemical formula C_12_H_12_O_5_; therefore, (*S*)-butin and its precursor, butein, are isomers. The isomer pair have been reported to coexist in wattle heartwood [12] and other plants [13,14,15]. This suggests the possibility of butein-to-(*S*)-butin transformation via plant metabolism, and supports the design of this study. This comparative study provides direct evidence for the structural optimization of natural phenolics as ferroptosis inhibitors.

In this study, we used erastin to induce ferroptosis in bone marrow-derived mesenchymal stem cells (BMSCs), which are considered important seed cells to treat degenerative diseases, such as geriatric diseases [16], by transplantation engineering. These have enhanced the clinical characteristics of the study.

In addition to comparing anti-ferroptosis bioactivities for butein and (*S*)-butin, anti-ferroptosis mechanisms for both will also be discussed. Recent research has indicated that ferroptosis regulation can be affected by the cellular redox environment [17]; anti-ferroptosis was closely associated with radical-trapping antioxidants [18,19,20] or even antioxidant chemical elements (e.g., selenium [21]). This suggests that the anti-ferroptosis mechanism may be mediated by antioxidants. Therefore, in this study, butein and (*S*)-butin were mixed with α,α-diphenyl-β-picrylhydrazyl radical (DPPH^•^), an antioxidant reaction probe, and the product mixture was further analyzed using ultra-performance liquid chromatography coupled with electrospray ionization quadrupole time-of-flight tandem mass spectrometry (UPLC-ESI-Q-TOF-MS). This cutting-edge approach provides precise *m/z* values and offers reliable information regarding the anti-ferroptosis mechanism of natural phenolics.

## 2. Results and Discussion

Erastin is a small molecule inducer of ferroptosis. It can inhibit glutathione peroxidase 4 (GPx4) and delete the corresponding GPx4 gene to cause glutathione depletion. As such, cellular LPO accumulates in phospholipids containing polyunsaturated fatty acids [22,23,24]. Cellular Fe^2+^ catalyzes the conversion of LPO to LOO^•^, which directly triggers cell death [5,25,26,27,28,29,30,31,32,33]. In fact, erastin has successfully induced ferroptosis in SH-SY5Y cells and tumor cells [6,20,21]. Therefore, erastin was added to BMSCs to induce ferroptosis in the study.

In the C11-BODIPY staining experiment, ferroptotic BMSCs in the erastin group displayed the most intense fluorescence, suggesting massive accumulation of LOO^•^ in the cellular membrane (Figure 2A) [34,35]. Correspondingly, the erastin group also had low viability (64.9%) and high total apoptosis percentages (35.1%) in the flow cytometric assay (Figure 2B).

However, in the butein and (*S*)-butin groups, fluorescence was less intense, indicating that butein and (*S*)-butin were effective in scavenging LOO^•^ radicals (Figure 2A). LOO^•^ radical scavenging should improve bmMSC viability. The flow cytometric assay revealed that the (*S*)-butin and butein groups possessed 74.02% and 85.44% viable cells, respectively (Figure 2B). The viability of butein (30 μM) was similar to that of non-oxidative dopamine (50 μM) [36]. Their ferroptotic inhibition towards BMSCs suggests that butein and (*S*)-butin are candidates for transplantation therapy for neurodegenerative diseases (such as Alzheimer’s disease) [28,29].

Since LPO affects ferroptosis, butein and its isomer (*S*)-butin were investigated for LPO inhibition using a linoleic acid system. As shown in Appendix A, the two isomers increased the LPO inhibition percentages with increasing concentration. Both the inhibition and formation of LPO are reported to stem from electron-transfer (ET) reactions [37,38,39,40,41]. To explore ET, the two were further investigated using the Fe^3+^-reducing antioxidant power (FRAP) assay and Cu^2+^-reducing antioxidant power (CUPRAC) assay. As shown in Appendix A, a direct correlation was found between the dose of butein or (*S*)-butin and the relative FRAP and CUPRAC percentages, indicating that both butein and (*S*)-butin had ET antioxidant potentials [42,43,44].

Several reports have suggested that ET is always accompanied by H^+^-transfer (proton transfer) in biological systems [45,46,47,48]. To evaluate proton transfer with butein and (*S*)-butin, we used a PTIO^•^ scavenging assay established by our team [49]. In the assay, varying IC_50_ values (the concentration with 50% radical inhibition or relative reducing power) were observed for the two isomers at different pH values (4.5, 6.0, and 7.4, shown in Table 1, and Appendix A). Such a pH effect indicates that, in aqueous solution, PTIO^•^ radical scavenging could involve H^+^ (i.e., proton). Thus, there was a proton transfer (PT) pathway. The involvement of the PT pathway can be attributed to a fact that phenolics always exhibit weak acidity. In fact, (*S*)-butin had a *p*Ka value of 7.11 at 3′-OH [50,51]. On the other hand, the PTIO^•^ scavenging reaction in pH 4.5 aqueous solution is indicated as an ET process [49]. The effectiveness of the two isomers at pH 4.5 implies that they also had the potential for ET. In short, the antioxidant process of butein and (*S*)-butin is considered to be a single process comprising ET accompanied by PT (designated as ET *plus* PT).

One important type of ET *plus* PT is the HAT (hydrogen atom transfer) mechanism [48,52,53,54,55,56,57], which was determined using DPPH^•^ as a probe. The DPPH^•^ probe, however, was recently suggested to have some limitations [35]. Nevertheless, DPPH^•^ determination has successfully accumulated a great deal of evidence in computational [58,59] and experimental models [60,61,62]. More importantly, two tested-samples butein and (*S*)-butin could not be dissolved in aqueous solution. Thus, we preferred to incubate DPPH^•^ with butein or (*S*)-butin in methanol solution. The product mixture was then measured using UPLC-ESI-Q-TOF-MS. As shown in Figure 3, butein incubated with DPPH^•^ gave a molecular ion peak at [M] = *m/z* 542.1174. This *m/z* value (542.1174) of the molecular ion peak is double the molecular mass of butein (272.0685) minus the relative mass of two hydrogen atoms. The experimental value of two hydrogen atoms (2.0124) had only 0.16% relative deviation from the calculated value (2.01565) [30], suggesting that there were two HAT reactions.

On the basis of these experimental data and previous reports [62,63], a HAT reaction may firstly occur at the 4-OH of butein (Figure 4A) [42,64]. Via the HAT reaction, butein is transformed to the butein-4-*O*-radical. A hydrogen atom is abstracted from the butein-4-*O*-radical by excess DPPH^•^ to form butein quinone (Figure 4A). Finally, the catechol moiety is easily oxidized to quinone [65,66,67,68,69].

However, some of the butein-4-*O*-radicals can produce a butein dimer, through radical resonance, radical adduct formation, and subsequent keto-enol tautomerization [8] (Figure 5A). This dimer gives rise to the aforementioned molecular peak (*m/z* 542.1174) and several fragment peaks in the mass spectrum. These MS peaks are accurately elucidated in Figure 5B, using time-of-flight tandem mass spectrometry. For example, the relative deviations of the CO loss between the experimental value of 27.995 (*m/z* 253.0509 − 225.0559) and the calculated value of 27.994915 were 3.0 × 10^−6^ (Figure 3C) [70]. These accurate and reliable analyses clearly demonstrate the butein dimer as one of antioxidant products of butein. The dimer formation supports the HAT mechanism for the antioxidant process, similar to reports from the literature [55,71,72].

Similar to butein, (*S*)-butin was also observed to yield a dimer by UPLC-ESI-Q-TOF-MS analysis (Figure 2). Its dimer was hypothesized to form via a 5′,5′-linkage. This (*S*)-butin 5′,5′-dimer tended to undergo retro-Diels–Alder fragmenting during MS, similar to other flavonoid dimers [63,73]. As a result, the (*S*)-butin 5′,5′-dimer generated characteristic MS peaks, such as *m/z* 389, 269, 253, and 151 (Figure 6).

Interestingly, when butein, (*S*)-butin, and DPPH^•^ were incubated together, UPLC-ESI-Q-TOF-MS analysis detected a new dimer with different chromatographic and MS peaks than those found for the butein 5,5-dimer or (*S*)-butin 5′,5′-dimer (Figure 3G). Using the HAT mechanism of DPPH^•^-scavenging and Pauling’s resonance theory, we hypothesized that this new dimer was a butein-(*S*)-butin 5,5′-dimer linked between the butein 5-position and the (*S*)-butin 5′-position (Figure 7A). To distinguish this dimer from biochemical heterodimers [74], we designated it as the butein-(*S*)-butin cross dimer in this study. MS displays evidence of the cross dimer, with peaks at *m/z* 431, 419, 413, 375, and 109 (Figure 3I and Figure 7B).

The evidence of dimers (especially the cross dimer) explains why (*S*)-butin and the (*S*)-butin dimer can coexist in *Cotinus coggygria* wood [15], and why fisetin and (±)-catechin coexist with their cross dimers fisetinidol-(4*α*→8)-(+)-catechin and epifisetinidol-(4*β*→8)-(+)-catechin in the same plant [15]. Other examples of cross dimers may also be explained using the above evidence, including the chalcone-isoflavone dimer [75], chalcone-flavonone heterodimer [76], chalcone-tannin hybrid [77], pyranomalvidin-procyanidin dimer [78], and anthocyanin-flavone dimer [79] (Appendix A). The coexistence of cross dimers may be the result of reactive oxygen species (ROS)-mediated metabolism, where a phenolic radical intermediate covalently combines with another phenolic radical intermediate to give cross dimer. The ROS may originate from metabolism, visible light [80], and even air. For heat-stressed plants, ROS may also come from ferroptosis [27,81]. Importantly, the above findings can also explain dityrosine-based protein–protein crosslinking and α-synuclein crosslinking in animal cells [82,83], since tyrosine is a phenolic amino acid and animal metabolism also involves ROS.

We established that both butein and (*S*)-butin can inhibit ferroptosis, possibly through antioxidant mechanisms, particularly the HAT mechanism. During the HAT-mediated antioxidant process, butein and (*S*)-butin can produce three types of dimeric products: The butein 5,5-dimer, (*S*)-butin 5′,5′-dimer, and butein-(*S*)-butin cross dimer. The relative anti-ferroptotic bioactivities of butein and (*S*)-butin differed. As seen in Figure 2A, butein exhibited less fluorescence than (*S*)-butin, indicating that butein more effectively prevented LPO accumulation in the cellular membrane than (*S*)-butin. Accordingly, butein displayed higher cellular viability and lower apoptosis than (*S*)-butin in the flow cytometry assay (Figure 2B).

A similar difference was also observed regarding the antioxidant bioactivities between butein and (*S*)-butin. As seen in Appendix A, both butein and (*S*)-butin dose-dependently increased the percentages in all five antioxidant assays, including linoleic acid emulsion assay, CUPRAC assay, Fe^3+^-reducing assay, PTIO^•^ radical-trapping assay, and DPPH^•^ radical-trapping assay. Linoleic acid however was one of polyunsaturated fatty acids and has been recently suggested as one target of ferroptosis [84,85]. The effectiveness of butein and (*S*)-butin in linoleic acid emulsion assay means that the anti-ferroptosis effect of butein and (*S*)-butin may be related to the LPO-inhibition on linoleic acid. However, as seen in Table 1, butein had lower IC_50_ values than (*S*)-butin in the five antioxidant assays. This indicates that, butein is a stronger LPO-inhibitor or antioxidant than (*S*)-butin. That their relative anti-ferroptotic activities parallel their relative antioxidant (or LPO-inhibitory) levels further supports the above hypothesis, namely, that the anti-ferroptotic action of butein and (*S*)-butin occurs via antioxidant mechanisms.

The difference in their activities is likely associated with CHI-mediated isomerization. As shown in Figure 1, isomerization from butein to (*S*)-butin is a cyclization reaction. Cyclization results in loss of the 2′-OH and thus lowers the antioxidant level [69,86]. More importantly, cyclization also causes saturation of the α,β-double bond. After the α,β-double bond is saturated, the β-carbon has an sp^3^-hybridized tetrahedral configuration. As such, the molecule is no longer planar (Figure 8). Loss of the exocyclic double bond and loss of molecular planarity reduces the region of π-π conjugation. From resonance theory, saturation of the exocyclic α,β-double bond prevents stabilization of the free radical intermediate after the HAT reaction [62]. We previously demonstrated that these changes greatly decrease the antioxidant level [87,88].

## 3. Materials and Methods

### 3.1. Chemicals, Animals, and Biological Kits

(*S*)-Butin (C_15_H_12_O_5_, CAS number: 492-14-8, M.W: 272.256, purity 97%, Appendix A); butein (C_15_H_12_O_5_, CAS number: 487-52-5, M.W: 272.256, purity 99%, Appendix A). 2,4,6-Tripyridyl triazine (TPTZ), (±)-6-Hydroxyl-2,5,7,8-tetramethylchromane-2-carboxylic acid (Trolox), 2,9-dimethyl-1,10-phenanthroline (neocuproine), and linoleic acid were obtained from Sigma-Aldrich (Shanghai, China). l-Ascorbic acid was obtained from J&K Scientific (Beijing, China). α,α-Diphenyl-β-picrylhydrazyl radical (DPPH^•^, C_18_H_12_N_5_O_6_) was obtained from Aladdin Chemical Ltd. (Shanghai, China). The 2-phenyl-4,4,5,5-tetramethylimidazoline-1-oxyl-3-oxide radical (PTIO^•^) was obtained from TCI Chemical Co. (Shanghai, China). Methanol and the other reagents were purchased from Guangdong Guanghua Chemical Plants Co., Ltd. (Shantou, China).

Sprague-Dawley rats (4 weeks, Identification code: 2018030714) were purchased from the Animal Centre of Guangzhou University of Chinese Medicine. Procurement, maintenance, and treatment of the animals were performed under the supervision of the Institutional Animal Ethics Committee in Guangzhou University of Chinese Medicine. Fetal bovine serum (FBS) and trypsin were from Gibco (Grand Island, NY, USA). The probe C11-BODIPY was purchased from Molecular Probes (Eugene, OR, USA). An annexin V/propidium iodide (PI) assay kit was purchased from BD Biosciences (Invitrogen, Carlsbad, CA, USA). Erastin was from MedChemExpress (Monmouth Junction, Middlesex County, NJ, USA). The complete medium with glucose for SD rat bone marrow mesenchymal stem cells was purchased from Cyagen Biosciences (Guangzhou, China). 

### 3.2. Extraction and Culture of Bone Marrow-Derived Mesenchymal Stem Cells (BMSCs)

The bone marrow-derived mesenchymal stem cells (BMSCs) were extracted and cultured using our routine experimental protocols [89]. Briefly, male Sprague-Dawley rats were collected, and the adherent soft tissues were removed. Both ends of the bones were cut away from the diaphysis with bone scissors. The bone marrow plugs were hydrostatically expelled from the bones by insertion of needles fastened to 10-mL syringes filled with complete medium; the needles were inserted into the distal ends of femora and proximal ends of the tibiae, and the marrow plugs expelled from the opposite ends. The cells were centrifuged and resuspended twice in complete medium; 5 × 10^7^ cells in 7–10 mL of complete medium were then introduced into 100-mm culture dishes. Two days later, the medium was changed and the nonadherent cells were discarded. The adherent cells were cultured in SD rat bone marrow mesenchymal stem cell complete medium with glucose, supplemented with 10% (*v*/*v*) fetal bovine serum. The cultured cells were seeded and grouped to study the prevention of erastin-induced ferroptosis of butin and (*S*)-butein.

### 3.3. Prevention of Erastin-Induced Ferroptosis in BMSCs

The erastin-induced ferroptosis model of BMSCs was created based on the recent literature [19,20] with modifications. To measure the anti-ferroptosis bioactivities of butin and (*S*)-butein, three assays were applied in the study. The three assays referred to the C11-BODIPY assay and flow cytometric assay.

The C11-BODIPY assay was used to characterize the degree of lipid peroxidation and performed using the method [90,91]. In brief, the cultured BMSCs mentioned were seeded at 1 × 10^6^ cells per well into 12-well plates. After adherence for 24 h, BMSCs were divided into control, model, and sample groups. In the control group, BMSCs were incubated for 12 h in Stel Basal medium. In the model and sample groups, BMSCs were incubated in the presence of erastin (20 μM). After incubation for 12 h, the mixture of erastin and medium was removed. The BMSCs in the model group were incubated for 12 h in Stel Basal medium while BMSCs in the sample group were incubated for 12 h in Stel Basal medium with the indicated 30 μM sample concentrations. The incubated cells were determined using the fluorescent probe C11-BODIPY (Invitrogen, Molecular Probes). Cells were incubated for 30 min prior to analysis with C11-BODIPY (2.5 μM). Photos were taken under a fluorescence microscope.

The flow cytometric assay was conducted according to previous methods [92,93,94,95]. In brief, the cultured BMSCs (Section 3.2) were seeded at 1 × 10^6^ cells per well into 96-well plates. They were washed twice with cold PBS, and then cells were resuspended in 1 × Binding buffer at a concentration of 1 × 10^6^ cells/mL. Then, 100 μL of the solution (1 × 10^5^ cells) was transferred to a 5-mL culture tube, and 5 μL of FITC Annexin V and 5 μL PI was added. The cells were gently vortexed and incubated for 15 min at room temperature in the dark, and 400 mL of 1×Binding Buffer were added to each tube after adherence for 12 h. BMSCs were divided into control, model, and sample groups. The three groups were analyzed by flow cytometry within 1 h. Each sample test was repeated in three independent wells.

### 3.4. Linoleic Acid Emulsion Assay

The anti-lipid peroxidation effects of butin and (*S*)-butein were investigated using the linoleic acid emulsion assay [96]. Briefly, linoleic acid emulsion was prepared using linoleic acid and Tween 20. Then, 1.5 mL of linoleic acid emulsion were mixed with 0.15 mL of sample methanolic solution (0.4–2.0 mg/mL) and 0.35 mL of 30% ethanol (*v*/*v*). The reaction mixture (total 2 mL) was incubated at room temperature for 72 h. Then, 0.15 mL of the mixture were added to 3.65 mL 75% ethanol (*v*/*v*), 0.1 mL NH_4_SCN (30%, *w*/*w*), and 0.1 mL FeCl_2_ (0.02 M in 3.6% HCl). The resulting mixture was measured with a UV–Vis spectrophotometer (Unico 2600A, Shanghai, China) at 500 nm. The inhibition percentage was calculated by the equation:(1)$$Inhibition\%\;=\;\frac{A_{0}-A}{A_{0}}\;\times \;100\%,$$
where *A*_0_ is the absorbance of the control without sample, and *A* is the absorbance of the reaction mixture with sample.

### 3.5. CUPRAC Assay

The CUPRAC assay was adapted from the method proposed by Apak [97], with small modifications, as described by Tian [98]. Twelve microliters of CuSO_4_ solution (0.01 M) and 12 μL of ethanolic neocuproine solution (7.5 mM) were added to a 96-well plate and mixed with different concentrations of samples (0–2.0 μg/mL). The total volume was then adjusted to 100 μL with a CH_3_COONH_4_ buffer solution (0.1 M), and mixed again to homogenize the solution. The mixture was maintained at room temperature for 30 min, and the absorbance was measured at 450 nm on a microplate reader (Multiskan FC, Thermo Scientific, Shanghai, China). The relative reducing power of the sample was calculated as follows:(2)$$Relative\;reducing\;effect\%\;=\;\frac{A-A_{min}}{A_{\max-}A_{min}}\;\times \;100\%,$$
where *A*_max_ is the maximum absorbance, *A*_min_ is the minimum absorbance, and *A* is the absorbance of the sample.

### 3.6. FRAP Assay

The FRAP assay was adapted from Benzie and Strain FRAP [99]. Briefly, the FRAP reagent was freshly prepared by mixing 10 mM TPTZ, 20 mM FeCl_3_, and 0.25 M pH 3.6 acetate buffer at 1:1:10 (volume ratio). The test sample (x = 0–10 μL, 0.5 mg/mL) was added to (20 − x) μL of 95% ethanol followed by 80 μL of FRAP reagent. The absorbance was measured at 595 nm after a 30 min incubation at 37 °C using distilled water as the blank. The relative reducing power of the sample compared to the maximum absorbance was calculated using the formula of Section 3.5.

### 3.7. PTIO^•^ Radical-Trapping Assay

The PTIO^•^-scavenging spectrophotometer assay was conducted in accordance with our method [49,84]. In brief, the test sample solution (x = 0–10 μL, 0.5 mg/mL) was added to (20 − x) μL of methanol, followed by 80 μL of an aqueous PTIO^•^ solution. The aqueous PTIO^•^ solution was prepared using 0.1 mM of phosphate buffer/methanol (1/4, *v*/*v*) solution (pH 4.5, 6.0 and 7.4). The mixture was maintained at 37 °C for 2 h, and the absorbance was then measured at 560 nm using a microplate reader (Multiskan FC, Thermo Scientific, Shanghai, China). The PTIO^•^ inhibition percentage was calculated based on the formula of Section 3.5

### 3.8. DPPH^•^ Radical-Trapping Assay

The DPPH^•^ radical trapping was determined as previously described [100]. Briefly, 80 μL of DPPH^•^ solution (0.1 M) were mixed with methanolic sample solutions at the indicated concentration (x = 0–10 μL, 0.5 mg/mL). The total volume of mixture was adjusted to 100 μL, and maintained at room temperature for 1 min, and the absorbance was measured at 519 nm on a microplate reader. The percentage of DPPH^•^ scavenging activity was calculated using the equation of Section 3.4.

### 3.9. UPLC-ESI-Q-TOF-MS Analysis of RAF Products of Two Isomers Interacting with DPPH^•^

Butein (2 mg/mL) was subjected to a reaction with DPPH^•^ (5 mg/mL) under the previous conditions [101]. A methanol solution of butein was mixed with a methanolic DPPH^•^ solution (5 mg/mL) with a molar ratio of 2:1, and then incubated for 24 h at room temperature. The product mixture was passed through a 0.22-μm filter.

The filtrate was analyzed using the UPLC-ESI-Q-TOF-MS method [63]. In the UPLC-ESI-Q-TOF-MS analysis, a Phenomenex Luna C_18_ column (2.1 mm i.d. × 100 mm, 1.6 μm, Phenomenex Inc., Torrance, CA, USA) was used as the chromatographic column. The sample injection volume was 3 μL. The sample (reaction products) in the chromatographic column was eluted by a mobile phase at a flow rate of 0.2 mL/min. The mobile phase, however, consisted of a mixture of methanol (phase A) and 0.1% formic acid water (phase B). The proportion of phase A and phase B was adjusted using a gradient program: 0–2 min, maintain 30% B; 2–10 min, 30%–0% B; 10–12 min, 0%–30% B. The Q-TOF-MS analysis was performed on a Triple TOF 5600*^plus^* mass spectrometer (AB SCIEX, Framingham, MA, USA) equipped with an ESI source, which was run in the negative ionization mode. The scan range was set at 50–1500 Da. The system was run with the following parameters: Ion spray voltage, −4500 V; ion source heater temperature, 550 °C; curtain gas pressure (CUR, N_2_), 30 psi; nebulizing gas pressure (GS1, Air), 50 psi; Tis gas pressure (GS2, Air), 50 psi. The declustering potential (DP) was set at −100 V, whereas the collision energy (CE) was set at −45 V with a collision energy spread (CES) of 15 V. The above experiment was repeated using (*S*)-butin (2 mg/mL), instead of butein (2 mg/mL).

In the cross-dimerization reaction, butein (2 mg/mL), (*S*)-butin (2 mg/mL), and DPPH^•^ (5 mg/mL) were mixed with each other. The molar ratio of butein:(*S*)-butin:DPPH^•^ was 4:4:1. The reaction and determination conditions were based on the above description.

### 3.10. Preferential Conformation Analysis by Computational Chemistry and Molecular Weight Calculation

The preferential conformation was analyzed based on force fields by computational chemistry. In brief, the energy minimization of butein and (*S*)-butin were respectively calculated through molecular mechanics II (MM2) using the Chem3D Pro14.0 program (PerkinElmer, Waltham, MA, USA) [102,103].

The Q-TOF-MS analysis is characterized by highly accurate *m/z* values, particularly molecular weights. The molecular weight calculation based on the formula is vital for comparison with the *m/z* values from the Q-TOF-MS analysis. In the present study, the molecular weight calculations were conducted based on the accurate relative atomic masses. The relative atomic masses of C, H, O, and N were 12.0000, 1.007825, 15.994915, and 14.003074, respectively [70].

### 3.11. Statistical Analysis

The results were reported as the mean ± SD of three independent measurements; the IC_50_ values were calculated by linear regression analysis, and independent-sample T-tests were performed to compare the different groups [104]. A *p* value of less than 0.05 was considered statistically significant. The statistical analyses were performed using the SPSS software 17.0 (SPSS Inc., Chicago, IL, USA) for windows. All of the linear regression analyses described in this paper were processed using version 6.0 of the Origin professional software (OriginLab Corporation, Northampton, MA, USA).

## 4. Conclusions

2′-Hydroxy chalcone butein and its isomer dihydroflavone (*S*)-butin inhibit ferroptosis through antioxidant action. Antioxidant action occurs by the HAT mechanism, as evidenced by the presence of the butein dimer and (*S*)-butin dimer. Interestingly, butein and (*S*)-butin can interact with each other to produce a cross dimer. The isomers display different antioxidant or anti-ferroptotic levels; this discrepancy can be attributed to biocatalyzed isomerization, which results in the loss of a phenolic -OH and produces a saturated α,β-double bond, impairing π-π conjugation in (*S*)-butin.

## Figures and Tables

**Figure 1 molecules-25-00674-f001:**
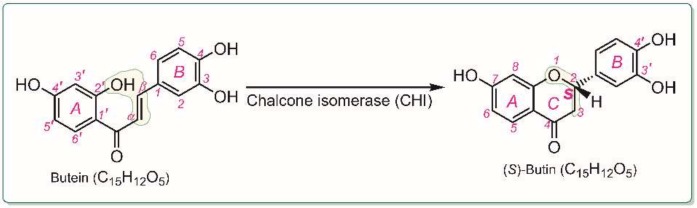
Possible isomerization of butein to (*S*)-butin, mediated by chalcone isomerase (CHI).

**Figure 2 molecules-25-00674-f002:**
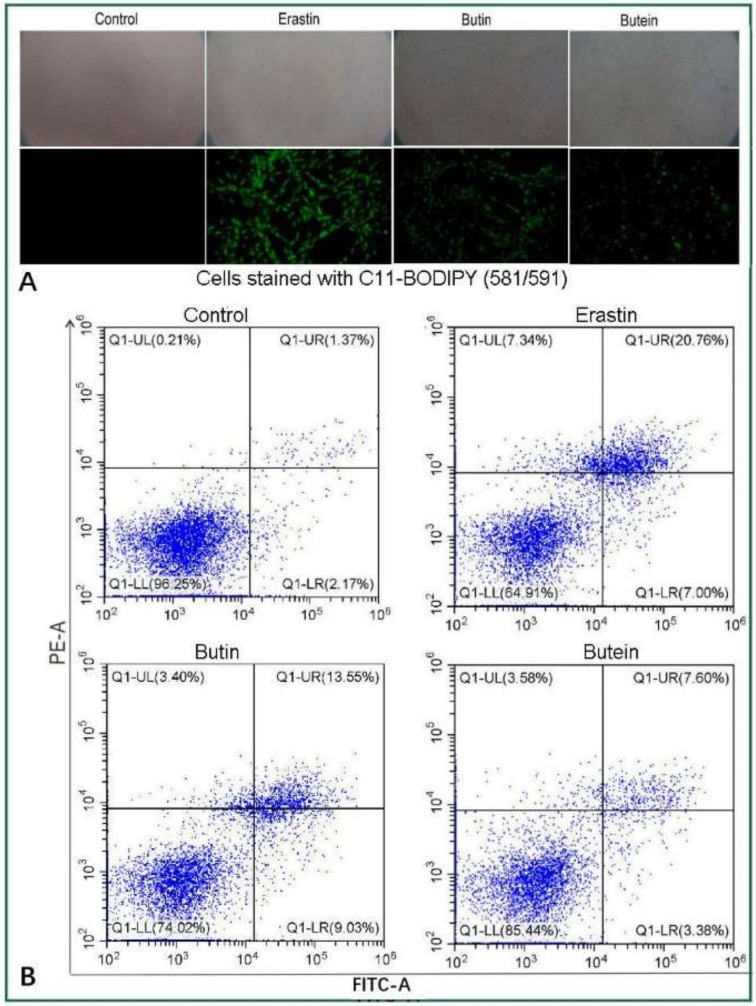
A Ferroptotic inhibition effects of butein and (*S*)-butin in bone marrow-derived mesenchymal stem cells (bmMSCs): (**A**) C11-BODIPY staining assay (concentrations of butein and (*S*)-butin were 30 μM); (**B**) flow cytometric assay (Q1, Q2, Q3, and Q4 showed the cellular death, late apoptosis, early apoptosis, and cellular viability, respectively; concentrations of butein and (*S*)-butin were 30 μM). FITC, fluorescein isothiocyanate.

**Figure 3 molecules-25-00674-f003:**
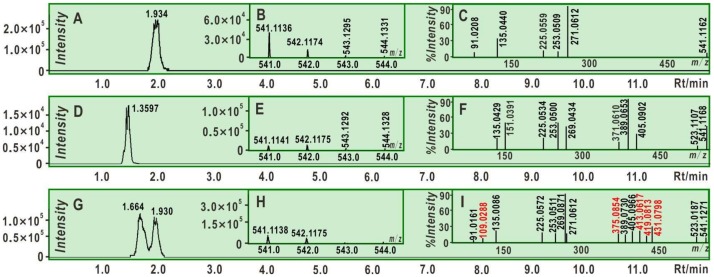
The ultra-performance liquid chromatography coupled with electrospray ionization quadrupole time-of-flight tandem mass spectrometry (UPLC-ESI-Q-TOF-MS) analysis for radical adduct formation (RAF) dimers of two isomers interacting with DPPH radical (α,α-diphenyl-β-picrylhydrazyl radical). (**A**) Chromatogram of the butein dimer when the formula [C_30_H_22_O_10_-H]^−^ was extracted; (**B**) primary MS spectra of the butein dimer (from Rt 1.8338 min peak); (**C**) secondary MS spectra of the butein dimer; (**D**) chromatogram of the (*S*)-butin dimer when the formula [C_30_H_22_O_10_-H]^−^ was extracted; (**E**) primary MS spectra of the (*S*)-butin dimer; (**F**) secondary MS spectra of the (*S*)-butin dimer; (**G**) chromatogram of the butein-(*S*)-butin cross dimer when the formula [C_30_H_22_O_10_-H]^−^ was extracted; (**H**) primary MS spectra of the butein-(*S*)-butin cross dimer (from Rt 1.664 min peak); (**I**) secondary MS spectra of the butein-(*S*)-butin dimer (the red means the specific peak of butein-(*S*)-butin dimer, distinguishing from butein dimer or (*S*)-butin dimer). Note: The UPLC-ESI-Q-TOF-MS analysis for standard butein and (*S*)-butin are listed in Appendix A.

**Figure 4 molecules-25-00674-f004:**
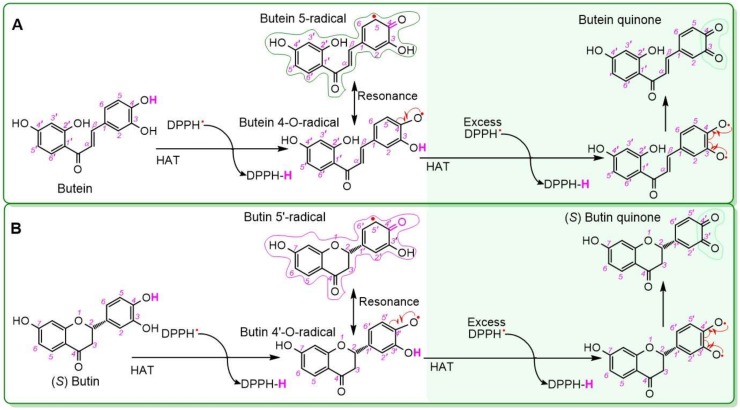
Hydrogen atom transfer (HAT) reactions of butein (**A**) and (*S*)-butin (**B**) with DPPH^•^ (the red curly single-barbed arrow indicates one electron transfer).

**Figure 5 molecules-25-00674-f005:**
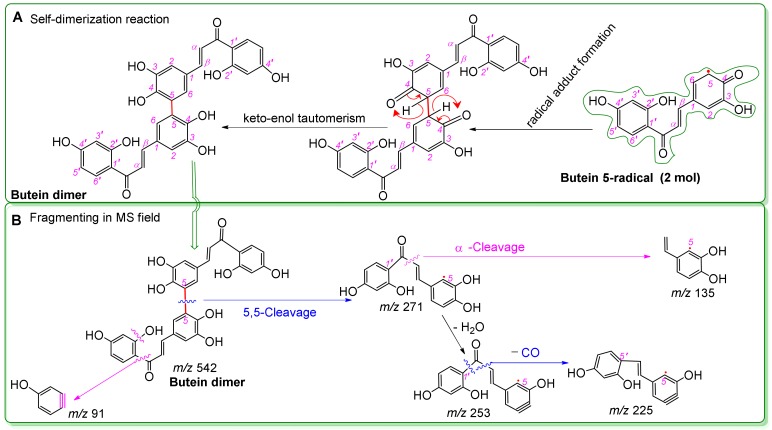
Proposed dimerization reaction of butein (**A**) and MS elucidations of butein dimer (**B**) (in Figure A, the red curly double-barbed arrow indicates a two-electron transfer; a curly arrow passes through an atom to indicate the position of new bond. In Figure B, MS was operated in negative ion mode. Accurate *m/z* values are shown in Figure 2 and are rounded to integers in MS elucidation. Other reasonable linking positions and cleavages should not be excluded in the MS elucidation).

**Figure 6 molecules-25-00674-f006:**
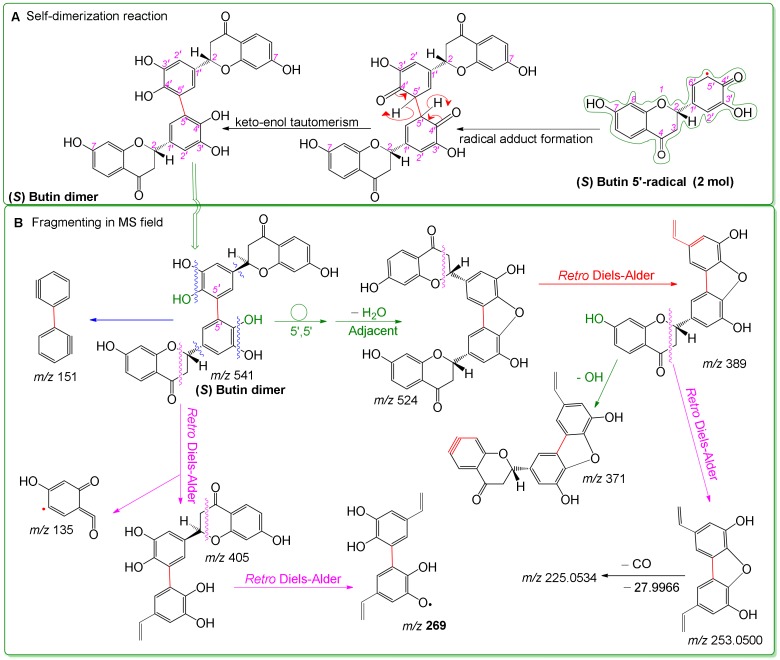
Proposed dimerization reaction of (*S*)-butin (**A**) and MS elucidations for the (*S*)-butin dimer (**B**). In Figure A, the curly double-barbed arrow indicates a two-electron transfer; a curly arrow passes through an atom to indicate the position of a new bond. For Figure B, MS was operated in negative ion mode. Accurate *m/z* values are shown in Figure 2 and are rounded to integers in MS elucidation. Other reasonable linking positions and cleavages should not be excluded in the MS elucidation).

**Figure 7 molecules-25-00674-f007:**
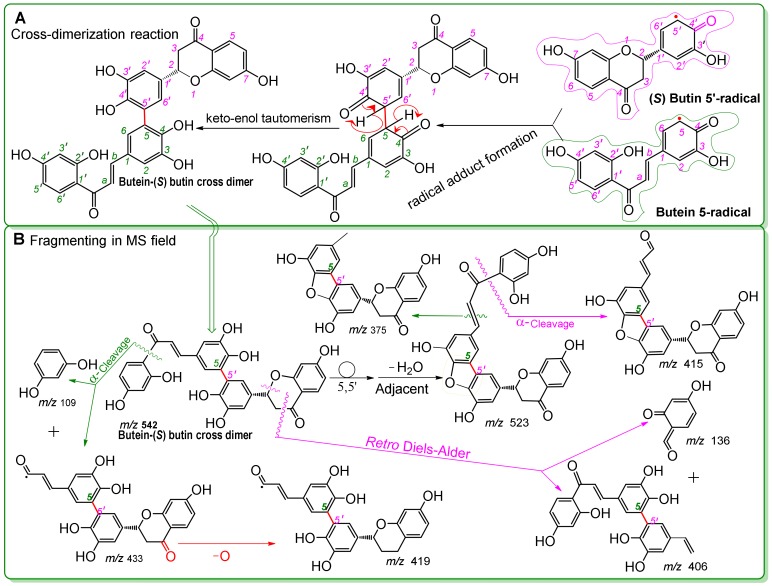
Proposed cross dimerization reaction of butein and (*S*)-butin (**A**) and MS elucidations for the butein-(*S*)-butin cross dimer (**B**) (in Figure A, the curly double-barbed arrow indicates a two-electron transfer; a curly arrow passes through an atom to indicate the position of a new bond. In Figure B, the MS was operated in negative ion mode. The circle indicates rotation of the 5,5′-σ bond. Accurate *m/z* values are shown in Figure 2 and were rounded to integers in MS elucidation. Other reasonable linking positions and cleavages should not be excluded in the MS elucidation).

**Figure 8 molecules-25-00674-f008:**
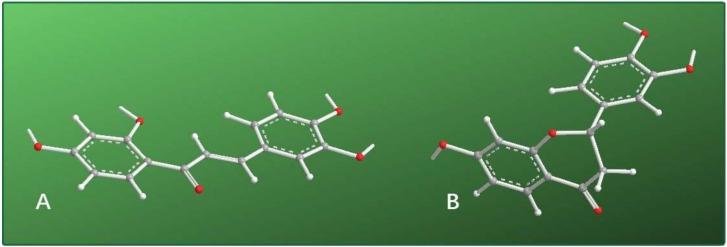
Molecular models of butein (**A**) and (*S*)-butin (**B**). The molecular model was created based on preferential conformations; the preferential conformation was analyzed using the Chem3D Pro14.0 program (PerkinElmer, Waltham, MA, USA).

**Table 1 molecules-25-00674-t001:** The IC_50_ values (μM) of two isomers in five colorimetric antioxidant assays.

Antioxidant Assays	Butein	(*S*)-Butin	Trolox	l-Ascorbic Acid
Linoleic acid emulsion	3.2 ± 1.4 ^a^	46.7 ± 7.5 ^b^	1.0± 1.5	N.D.
Cu^2+^-reducing	36.2 ± 0.1 ^a^	44.4 ± 1.9 ^b^	91.6 ± 3.8	83.2 ± 1.2
Fe^3+^-reducing	5.3 ± 0.1 ^a^	5.6 ± 0.1 ^b^	8.7 ± 0.2	4.5 ± 0.2
PTIO^•^-trapping pH 4.5	16.4 ± 0.9 ^a^	18.4 ± 0.4 ^b^	9.1 ± 0.2	8.2 ± 0.2
PTIO^•^-trapping pH 6.0	9.3 ± 0.9 ^a^	47.6 ± 7.1 ^b^	11.1 ± 0.5	8.3 ± 0.5
PTIO^•^-trapping pH 7.4	12.9 ± 0.4 ^a^	17.4 ± 0.4 ^b^	5.2 ± 2.8	4.7 ± 1.1
DPPH^•^-trapping	15.8 ± 0.5 ^a^	29.9 ± 0.5 ^b^	23.3 ± 0.7	27.0 ± 0.2

The IC_50_ value is defined as the concentration with 50% radical inhibition or relative reducing power, calculated by linear regression analysis, and expressed as the mean ± SD (*n* = 3). The linear regression was analyzed by using Origin 6.0 professional software. The IC_50_ values with different superscripts (a or b) among the two isomers are significantly different (*p* < 0.05). Trolox and l-Ascorbic Acid were used as the positive control. All dose-dependent curves are given in Appendix A. N.D., no detected.

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
