# Peer review of "Simultaneous Study of Anti-Ferroptosis and Antioxidant Mechanisms of Butein and (S)-Butin"

_molecules, 2020, doi:10.3390/molecules25030674_

Round 1

Reviewer 1 Report

Dear Authors,

The paper presents valuable information about antioxidant and anti-ferroptosis activity of butein and (S)-butin. Nevertheless some points should be improved:

Introduction: the part 'In the study, ferroptosis...... (...) characteristics of the study' should be included at the end of Introduction or reduced to minimum because the information apply to the methods which are included in other section. The section Results and Discussion should be divided into parts for all performed studies (the results for individual studies should be tighter stressed) In present form it is hard to read. 

Reviewer 2 Report

This manuscript is about the ability of two natural polyphenols (butein and butin) to counteract ferroptosis. The effect is explained (qualitatively) by showing that butein is a better radical quencher than butin. The topic is interesting, but the manuscript has many flaws that render it unsuitable for publication in the present form.

1) lack of positive control. Butein and butin have some effect on ferroptosis, but they should be compared to a reference compound that is known to block ferroptosis.

2) Pratt and co-workers have recently demonstrated that DPPH is a bad method to assess radical trapping activity linked to ferroptosis. Thi reference must be considered and discussed. https://doi.org/10.1016/j.chembiol.2019.09.007

3) It is not clear to me how authors demonstrate the position of the linkages in butein and butin dimers. Moreover, dimer formation strongly depends on reactant concentration. I suppose that this reaction is negligible under physiological conditions.

4) the sentences at lines 108-111 are not clear.

5) in figure 4: "excess" DPPH not "excessive"

6) In Table 1, entry: IC50 of butin with PTIO radical at pH 6. Why this value is so large respect pH 4.5 and 7.4?

Round 2

Reviewer 2 Report

The authors have improved the manuscript. Some minor points are reported below.

1) I understand that it is difficult to add now new experiments. However, authors may find other ways to compare their results with the data reported in the literature, for instance by comparing the % of viable cells.

3) If authors can not demonstrate the position of the link in the dimers, they should avoid mentioning only the 5 positions in the text. Lines 124-126 should be modified. Actually, in butein the radical can be placed also in the beta position (if placing the radical on the 4-OH).

Author Response

(1)The authors have improved the manuscript. Some minor points are reported below.I understand that it is difficult to add now new experiments. However, authors may find other ways to compare their results with the data reported in the literature, for instance by comparing the % of viable cells.

→ Now we have added a comparison with dopamine. Please kindly see: Line 88, 490. The addition of comparison has improved the quality of our manuscript. Thank you.

(2)If authors can not demonstrate the position of the link in the dimers, they should avoid mentioning only the 5 positions in the text. Lines 124-126 should be modified. Actually, in butein the radical can be placed also in the β-position (if placing the radical on the 4-OH).

→ I agree with the comment. Thus, I have revised the Line 125-126.